# Fish Rescue during Streamflow Intermittency May Not Be Effective for Conservation of Rio Grande Silvery Minnow

**Thomas P. Archdeacon** [1,*] **, Tracy A. Diver** [1] **and Justin K. Reale** [2]

1   U.S. Fish & Wildlife Service, New Mexico Fish & Wildlife Conservation Office, Albuquerque, NM 87131, USA; tracy_diver@fws.gov
2   U.S. Army Corps of Engineers, Albuquerque District, Albuquerque, NM 87131, USA; justin.k.reale@usace.army.mil
*   Correspondence: thomas_archdeacon@fws.gov

**Abstract:** Streamflow intermittency can reshape fish assemblages and present challenges to recovery of imperiled species. During streamflow intermittency, fish can be subjected to a variety of stressors, including exposure to crowding, high water temperatures, and low dissolved oxygen, resulting in sublethal effects or mortality. Rescue of fishes is often used as a conservation tool to mitigate the negative impacts of streamflow intermittency. The effectiveness of such actions is rarely evaluated. Here, we use multi-year water quality data collected from isolated pools during rescue of Rio Grande silvery minnow *Hybognathus amarus*, an endangered minnow. We examined seasonal and diel water quality patterns to determine if fishes are exposed to sublethal and critical water temperatures or dissolved oxygen concentrations during streamflow intermittency. Further, we determined survival of rescued Rio Grande silvery minnow for 3–5 weeks post-rescue. We found that isolated pool temperatures were much warmer (>40 °C in some pools) compared to upstream perennial flows, and had larger diel fluctuations, >10 °C compared to ~5 °C, and many pools had critically low dissolved oxygen concentrations. Survival of fish rescued from isolated pools during warmer months was <10%. Reactive conservation actions such as fish rescue are often costly, and in the case of Rio Grande silvery minnow, likely ineffective. Effective conservation of fishes threatened by streamflow intermittency should focus on restoring natural flow regimes that restore the natural processes under which fishes evolved.

**Keywords:** thermal stress; salvage; climate change; drought; hypoxia

## 1. Introduction

Intermittency is a common and natural condition for many of the world's streams [1]. However, many regions of the world are predicted to experience decreased precipitation, altering historical streamflow regimes [2]. Currently, climate change coupled with human-mediated water abstraction has already increased the frequency of streamflow intermittency in some areas [3,4], threatening endemic fishes [5]. These species now face increasing water temperatures and risk of stranding in isolated pools [6–8]. Some stream temperatures are projected to exceed the thermal critical maxima for many fishes [9], which may lead to temperature-dependent mortality, population declines, or other sublethal effects [10–12]. Exposure to intermittent habitats and elevated water temperatures can reshape fish assemblages [13,14], which may result in shifts towards more homogenous assemblages dominated by extremophile species [15]. Understanding the long-term consequences of drought on fish communities, including the effects of water temperature and streamflow intermittency, will likely be an ever-increasing challenge for native fish conservation [16,17].

Streamflow intermittency can directly impact fish communities when historically perennial systems become disconnected, forcing individuals to seek refuge in more contracted and often less hospitable habitats [14,15,18]. The Middle Rio Grande (MRG) covers ~330 km through central New Mexico, USA (MRG; Figure 1). This reach is affected by seasonal and supra-seasonal drought that often results in streamflow intermittency. Historically, the MRG was a large, wide, and shallow river dominated by sandy substrates with considerable intra-annual variation in flows with peak runoff driven by spring snowmelt and low summer flows [19]. Historical accounts of streamflow intermittency prior to the 1890s occurred, but intermittency was not observed annually [19]. Low-flows and intermittency became more common in the MRG in the 1900s as more water management infrastructure was constructed in New Mexico and Colorado [19]. Indeed, the MRG fish assemblage formerly included species like shovelnose sturgeon *Scaphirynchus platorhynchus* and American eel *Anguilla rostrata* that would be intolerant of frequent drying [20]. More recently, frequent supra-seasonal drought [21], declines in snowpack [22,23], and human-mediated water abstraction of up to 95% [24,25] in the MRG Basin have resulted in long periods of summer streamflow intermittency, e.g., >100 days and >80 km in extreme years, averaging around 38 days and 35 km annually [18].

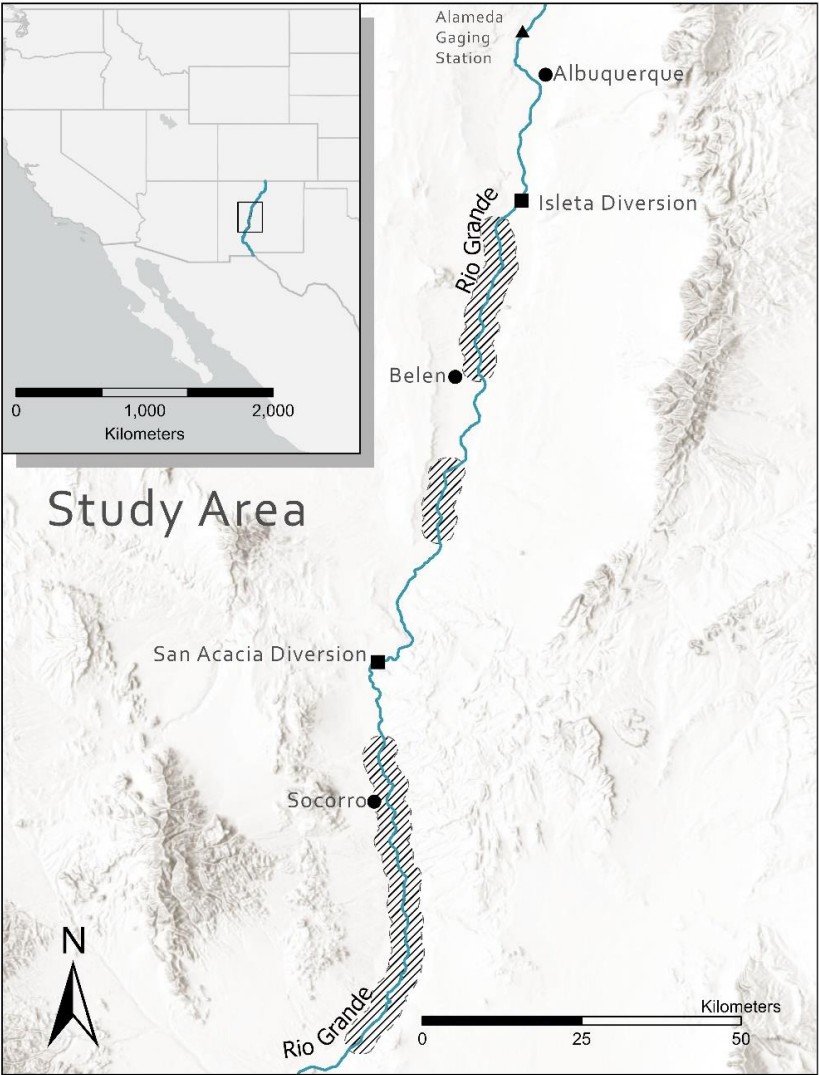

**Figure 1.** Areas of streamflow intermittency (crosshatched areas) in the Middle Rio Grande Basin of New Mexico, where temperatures of isolated pools were collected from 2011 to 2016. Circles represent metropolitan areas, squares are surface flow diversion structures, and the triangle is a streamflow and water quality gauging station.

The MRG is designated as a critical habitat for the only remaining wild population of the endangered Rio Grande silvery minnow *Hybognathus amarus*, a small-bodied member of the family Cyprinidae [26,27]. Rio Grande silvery minnow were once widespread throughout the mainstem Rio Grande and Pecos Rivers, from north central New Mexico to the Texas coast, but now occurs only in the MRG [28,29]. Rio Grande silvery minnow exhibits an opportunistic life-history strategy [30]. Fishes with this life-history strategy are typically small, short-lived, have high reproductive output, and maintain high demographic resilience [31]. Rio Grande silvery minnow have a relatively short lifespan, with few living > 2 years [32]; they reach maturity in the first year of life [33] and have relatively high fecundity for their body size [34]. Rio Grande silvery minnow spawn non-adhesive, neutrally buoyant ova directly into the water column [35], a mode of reproduction found in several other cyprinid species in the Great Plains of the western United States [36]. For Rio Grande silvery minnow, annual reproductive success is highly variable and is tied to high flows in spring [18,33,37], likely through creation of nursery habitats for larvae and juveniles [38,39]. Rio Grande silvery minnow were listed as endangered in 1994. Prior to Rio Grande silvery minnow being listed as endangered, four other species of minnow with similar reproductive biology were extirpated from the MRG during the 20th century [28].

Rio Grande silvery minnow are directly impacted by intermittent streamflow. During periods of river drying, Rio Grande silvery minnow are forced into isolated pools [37]. Compared to other intermittent streams in the American Great Plains region [15,40] and around the world [41], isolated pools in the MRG are shallow (<0.6 m) and short-lived, with only a small percentage persisting > 4 days [37]. Globally, isolated pools can last for weeks to years and allow the persistence of fishes and other organisms [42–44]. Isolated pools that support fish can be important for fishes that inhabit seasonally intermittent streams by providing habitat patches that reduce mortality compared to areas with no surface water [45]. However, in parts of the MRG, the river channel is perched above irrigation canals built for downstream water deliveries [46], which has resulted in rapid onset of intermittent conditions and evaporation of isolated pools due to a lack of connection to groundwater. In the absence of fish rescue, stranding in isolated pools in the MRG would result in almost certain mortality of fishes during summer months [37]. Thus, immediate rescue of Rio Grande silvery minnow, i.e., collection from isolated pools and translocation to areas with perennial surface flow, is performed before complete desiccation of isolated pools occurs as a conservation action to mitigate for the negative effects of streamflow intermittency.

Fish rescue and translocation are commonly used for mitigating the negative effects of stranding and streamflow intermittency [8,47,48]. However, rescue efforts are costly and rarely evaluated for effectiveness [8]. Exposure to stressors prior to translocation, as well as capture and transport stress, may limit survival after rescue. Evaluation of rescue efforts has focused on salmonids and the economic cost–benefit of translocating stranded fish [49,50]. Benefits to the population, simulated or realized, require that rescued fish survive to reproduce [50]; however, this assumption is likely both species and season dependent. Within the MRG, rescue and translocation of Rio Grande silvery minnow stranded during streamflow intermittency has been used as a management action for conservation since the early 2000s [18]. Fish are rescued and transported to areas with perennial flow each day after intermittency begins. However, the short-term survival of these fish after rescue and, ultimately, the effectiveness of rescue for conservation of the species is unknown. Many factors may decrease the effectiveness of fish rescue in the MRG. Pools may experience extreme temperature or dissolved oxygen fluctuations during streamflow intermittency in summer months [51]. Even though fish are typically rescued and translocated within 24 h of being stranded, they are likely exposed to high, and possibly lethal, temperatures, hypoxic conditions, and other stressors, such as crowding and predation prior to rescue. All of these factors may reduce the short-term survival of Rio Grande silvery minnow rescued from isolated pools.

Acute temperature and dissolved oxygen tolerances for larval, juvenile, and adult Rio Grande silvery minnow have been quantified in a laboratory setting [52]. Temperature tolerance was determined

through the incipient lethal temperature technique [52]. This method requires an abrupt transfer of fish to temperatures above or below the acclimation temperature, and lethal thresholds are determined when 50% mortality occurs within a set timeframe [53–55]. Acute dissolved oxygen tolerance was determined similarly [52]. For adult Rio Grande silvery minnow, the 50% lethal temperature threshold was 32.8 °C within a 24-h period; however, mortality began to occur at temperatures > 30 °C. Juveniles had higher 50% lethal temperature thresholds of 36.7 °C, with mortality beginning at about 34.5 °C within a 24-h period. For dissolved oxygen, juveniles experienced 50% mortality at 0.7 mg/L with no mortality at concentrations > 1.9 mg/L. For adults, these limits were 0.8 mg/L for 50% mortality and 1.1 mg/L for no acute mortality. While these types of tests can be critical for defining critical limits, the effects of chronic exposure to sublethal or fluctuating water temperatures or hypoxic conditions remains largely unknown. Chronic exposure to high, sublethal temperatures can affect metabolic rates [56], slow growth, and reproduction, and can alter behavior and increase stress and susceptibility to disease in other fishes [12]. Chronic hypoxia can also cause stress, affect swimming performance and metabolic functions [57], and result in transgenerational reproductive impairment [58]. Thus, understanding both the risk and effects of exposure to poor water quality may help managers determine the value of fish rescue and improve our understanding of the effects of rising water temperatures on fish assemblages [16].

Rescue efforts remain a major management strategy for Rio Grande silvery minnow during periods of streamflow intermittency and have been employed every year since 2009 [37]. In order to evaluate the severity of thermal conditions fish are exposed to prior to rescue, we used multi-year data collected during fish rescue to estimate hourly and seasonal changes in pool temperatures during streamflow intermittency. We compared this to the thermal regime of an upstream perennial reach. We also examined the number of Rio Grande silvery minnow exposed to elevated water temperatures or hypoxic conditions prior to rescue to assess the extent to which individuals in this endangered population are subjected to damaging water quality conditions. Finally, we estimated post-rescue survival of Rio Grande silvery minnow collected from isolated pools formed during streamflow intermittency as a preliminary evaluation of the effectiveness of fish rescue. Our results will help provide realistic estimates of the water quality conditions fishes are exposed to during streamflow intermittency, which will inform future studies examining lethal and sublethal effects of water temperature and hypoxia. Further, our results will inform conservation actions aimed at mitigating the effects of streamflow intermittency and stranding.

## 2. Materials and Methods

During irrigation season, March through November, surface water is diverted at multiple diversion dams within the MRG. During warm summer months, surface flow diversions can result in streamflow intermittency in areas below the Isleta and San Acacia Diversion dams (Figure 1). Typically, flows are decreased over several days and constant bypass is held at a diversion dam. Intermittent sections expand over these days, then remain relatively constant until further streamflow diversion is needed, reducing flows, or precipitation increases flows. Intermittency occurs in a mid-reach pattern, with perennial areas upstream and downstream of drying [59]. The pattern of drying is due to upstream diversions, irrigation return flows, channel perching, and groundwater influences. Multiple drying and wetting cycles can occur annually due to monsoon rain events or irrigation demands [18]. Observations of the channel conditions are made daily during this period of time; crews are dispatched to rescue fish when new areas of isolated pools form. The amount of newly dewatered channel is variable, ranging from 0.1 km up to 8 km [37], resulting in up to 130 new isolated pools per day. Fish rescue occurred within one day of streamflow intermittency, usually the morning through afternoon after isolated pools formed. During our study period from 2011 to 2016, we visited all isolated pools that occurred due to streamflow diversions. We recorded counts of Rio Grande silvery minnow collected in each pool, dissolved oxygen, temperature, maximum depth (0.1 m), date, and time of day. Fishes were sampled using 3.0 m by 1.0 m seines (3.2 mm mesh size). All Rio Grande silvery minnow collected

were transported in ~130-L fiberglass tanks mounted on all-terrain vehicles. We supplied pure oxygen to transport tanks to maintain ~100% saturation. Fish were released the same day in nearby areas with surface flows that were not expected to become intermittent. Depending on the number of pools that formed, fish spent 1 to 6 h in transport tanks before being released.

We measured water temperature (±0.01 °C) and dissolved oxygen (±0.01 mg L$^{-1}$) of all pools with a multi-parameter probe (YSI 556 MPS; Yellow Springs Institute) where maximum depth was recorded at the time of fish rescue. We were not able to collect temperature or dissolved oxygen at every pool due to equipment malfunction or data recording errors. We assumed isolated pools were thermally unstratified because most were <0.60 m in depth [37]. Previous studies support this assumption: in 2007, continuous (e.g., 15-min intervals) water quality monitoring conducted in five isolated pools over several days showed less than a 1 °C difference between surface temperature and temperature above the substrate [60].

We set a range of water temperature criteria based on preliminary laboratory data for Rio Grande silvery minnow [52] and similar species found in North America [55,61]. Based on these studies, we classified temperatures into four broad categories: <30 °C—no adverse effects, 30–33 °C—minimal adverse effects, including loss of equilibrium and muscle spasms, >33–36 °C—lethal for adults and sublethal for juveniles, and >36 °C—lethal for all age classes. These broad categories represent a continuum of possible adverse effects, but chronic, sublethal effects of elevated water temperature on Rio Grande silvery minnow are unknown. In order to place these effect thresholds into context with water temperatures experienced during flowing conditions, we compiled a 15-min interval temperature record over the same time period on the MRG at Alameda Gauging Station (Figure 1) from two locations c. 400 m from one another. Specifically, records collected by the University of New Mexico [62] and the U.S. Geological Survey (Gage No. 08329918) were combined to minimize temporal gaps. This location is within the occupied Rio Grande silvery minnow critical habitat that also maintains perennial surface flow. Similarly, we defined broad categories for dissolved oxygen: >2.0 mg L$^{-1}$—no adverse effects, 1.0–1.9 mg L$^{-1}$—minimal mortality for adults but potentially lethal for juveniles, and <1.0 mg L$^{-1}$—potentially lethal for both adults and juveniles.

We estimated average pool temperatures or stream temperatures by date and time of day from 2011–2016. We used a linear mixed-effects model with a sinusoidal response. We used both first and second harmonic sine and cosine terms for time of day (period = 24) and ordinal day of year (period = 365) to model the cyclical annual and diel temperature fluctuations. We treated these as continuous fixed effects and year as a random effect to predict the mean temperature of isolated pools over time. We used the same model on each water temperature recorded at the Alameda Gauging Station and independently of isolated pools. Statistical models were run in the program R version 4.0.2 using the package "lme4" [63,64].

We also estimated post-rescue survival of Rio Grande silvery minnow. Rather than immediately returning fish to perennial areas, we retained smaller samples of Rio Grande silvery minnow from isolated pools and held them in sterile laboratory conditions for 5 to 7 weeks. Additional fish were collected from areas with surface flow to serve as control groups (3 replicates, 311 total fish) and compared to those collected in isolated pools (9 replicates, 2289 total fish) from March through August in 2018 and June through August in 2020 (Table 1). Fish spent approximately one hour longer in transport tanks—compared to those released directly in areas with surface flows—while being transported back to laboratory aquaria. Fish were acclimated to water in aquaria by slowly replacing tank water with aquarium water over 15–45 min, which was similar to fish released in areas with surface flow. Fish were held indoors in a 5100-L indoor recirculating system consisting of seven individual fiberglass tanks, one of which functioned as a sump and contained filtration and aeration equipment to maintain oxygen levels at 100% saturation. Each tank measured 180 × 75 × 60 cm. The system was filled with municipal water passed through a reverse-osmosis filter. Water temperature in tanks was not continuously monitored and fluctuated with ambient indoor air temperatures. However, discrete temperature measurements in holding tanks were between 19 and 23 °C. Water drained through a standpipe to a

sand filter and ultraviolet sterilizing filter. Flow to each tank was approximately 1000 L/h. This system has held a variety of fish species with minimal mortality of control groups [65,66]. We fed fish flake food specifically designed for Rio Grande silvery minnow [67] twice daily to satiation, and excess food and waste were siphoned from tanks twice per week for the duration of the experiment. We recorded mortality daily.

**Table 1.** Dates, stream conditions, number, and survival of Rio Grande silvery minnow rescued in the Middle Rio Grande, New Mexico.

| Date | Conditions | Number | Survival |
|---|---|---|---|
| 26 March 2018 | Continuous | 102 | 76.4 |
| 24 April 2018 | Continuous | 67 | 100 |
| 9 May 2018 | Continuous | 142 | 73.9 |
| 3 April 2018 | Intermittent | 250 | 24.4 |
| 10 April 2018 | Intermittent | 300 | 43.7 |
| 8 July 2018 | Intermittent | 226 | 2.6 |
| 4 August 2018 | Intermittent | 267 | 6.6 |
| 11 June 2020 | Intermittent | 45 | 8.9 |
| 12 June 2020 | Intermittent | 290 | 1.4 |
| 14 June 2020 | Intermittent | 101 | 8.9 |
| 14 July 2020 | Intermittent | 389 | 5.4 |
| 7 August 2020 | Intermittent | 421 | 5.0 |

## 3. Results

We sampled 7597 pools during the six-year study period. We measured temperature in 6780 pools and dissolved oxygen in 5555 pools. Pool temperatures at the time of rescue ranged from 6.4 to 41.0 °C. Dissolved oxygen ranged from 0.01 to 21.35 mg L$^{-1}$. Across all days, pools were hottest each day from ~13:00 to 18:00 (Figure 2). Isolated pools rarely exceeded effect levels of >30 °C prior to 1000 h. The model results (Table A1) show that the estimated mean water temperatures began exceeding 30 °C in mid-June and persisted through August (Figure 3), with the largest daily duration at or above this temperature occurring in August. The highest mean pool temperatures occurred in August between ~14:00 and 16:00 (Figure 3). Isolated pools had diel water temperature fluctuations of >10°C, with these pools cooling at a faster rate in evenings compared to rates of warming in the mornings (Figure 3). In the perennial flow reach, maximum estimated mean water temperatures were <28 °C and exhibited diel fluctuations of ~5 °C (Figure 3).

Dissolved oxygen was also variable among pools. Generally, dissolved oxygen in the pools decreased as temperature increased. However, values above and below the temperature-dissolved oxygen solubility curve [68] (Figure 2), suggesting other physical (e.g., groundwater contributions) and biological controls (i.e., gross primary productivity and ecosystem respiration), are influencing the dissolved oxygen among pools.

From 2011 to 2016, we rescued 32,951 Rio Grande silvery minnow from pools where water temperature was measured. The majority of fish were juveniles (63.9%). A similar number of both juveniles (N = 10,041) and adults (N = 10,251) were collected in pools that were <30 °C (5688 isolated pools), which represented the majority (61.6%) of rescued fish. We found a substantial portion (10,895 juveniles, 1179 adults) of Rio Grande silvery minnow in pools above the 'no adverse effects' threshold of >30 °C (643 isolated pools). Of those, 351 isolated pools had water temperatures > 33 °C and contained 85 juveniles and 311 adults. Ninety-nine isolated pools had lethal water temperatures ≥ 36 °C and contained four juveniles and 74 adults.

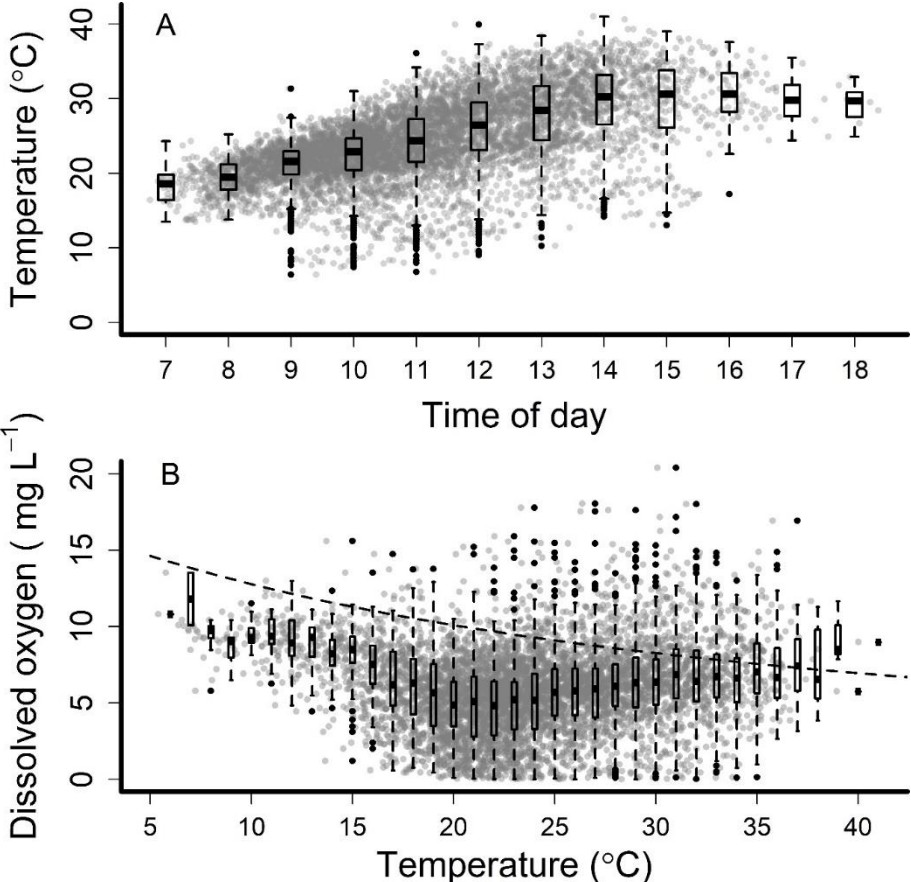

**Figure 2.** Water temperature of isolated pools by time of day (**A**) and dissolved oxygen by water temperature (**B**) that formed during streamflow intermittency in the Middle Rio Grande, New Mexico, June through October from 2011 to 2016. Individual pools are represented by gray dots and box-and-whisker plots depict the median (bar), interquartile range (box), points within 1.5 times the interquartile range (whiskers), and outliers (black dots). The dashed trendline represents the effect of temperature on solubility of oxygen in freshwater (chlorine and salinity = 0 ppm) at 760 mm Hg [68].

The majority of isolated pools (N = 5059) had dissolved oxygen concentrations ≥ 2.0 mg L$^{-1}$ and contained the majority of Rio Grande silvery minnow: 9960 juveniles and 3383 adults. Fewer isolated pools had dissolved oxygen concentrations between 1.0 and 2.0 mg L$^{-1}$ (N = 272) and had 364 juveniles and 1051 adults. Finally, we found 224 isolated pools with <1.0 mg L$^{-1}$ of dissolved oxygen and containing three juveniles and 323 adults.

Survival of Rio Grande silvery minnow rescued from isolated pools was generally poor. Fish rescued during surface flow conditions early in the year had higher survival than those rescued during streamflow intermittency, ≥74% compared to <50% (Figure 4). Further, among groups rescued from isolated pools, those collected earlier in the year had 3 to 20 times higher survival compared to those rescued in June and later (Table 1).

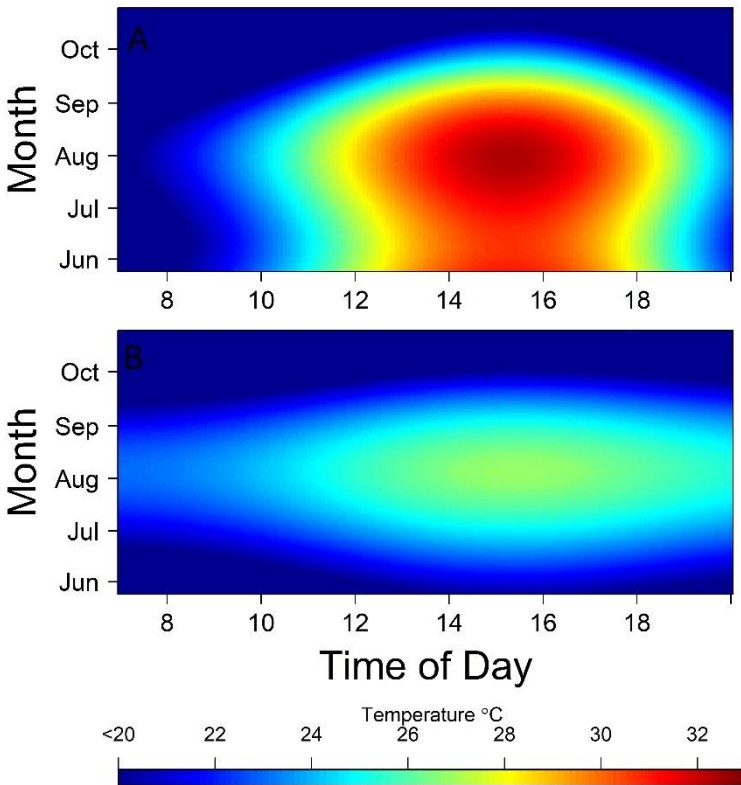

**Figure 3.** Heat map showing estimated mean temperature of isolated pools (**A**) or areas with perennial flow (**B**) from June through October, 2011 to 2016 in the Middle Rio Grande, New Mexico, by time of day and date, with cooler temperatures shown in blue.

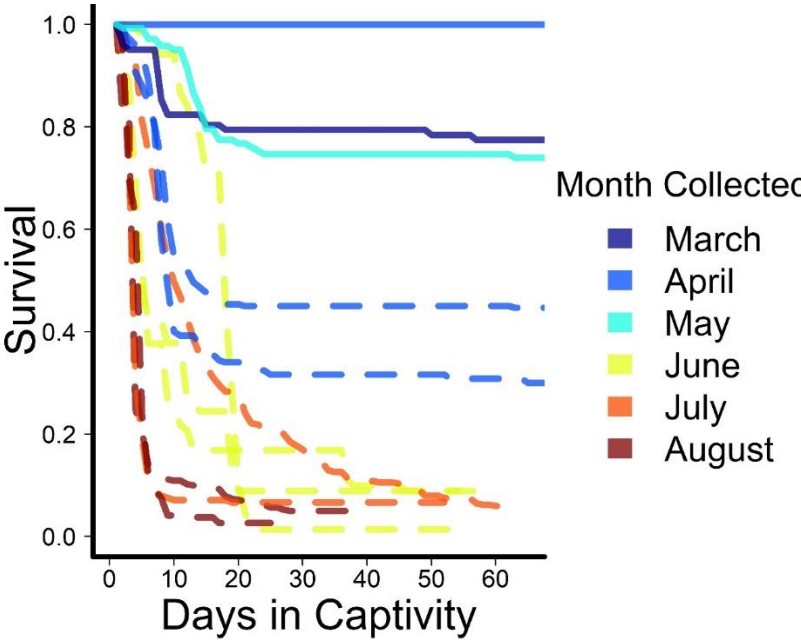

**Figure 4.** Daily cumulative survival of Rio Grande silvery minnow captured during surface flows (solid lines) or during streamflow intermittency (dashed lines) and held in captivity in 2018 and 2020.

## 4. Discussion

As expected, we found a substantial effect of time of day and ordinal day on pool temperature, with the hottest recorded water temperatures occurring in the late afternoon of June through August. Pool temperatures regularly exceeded levels found in an upstream section of the MRG that maintains continuous streamflow, suggesting that Rio Grande silvery minnow are exposed to unusually high temperatures during streamflow intermittency. Indeed, the maximum water temperature we recorded from isolated pools was >10 °C higher than the maximum temperature recorded in the connected upstream reach. Although these water temperature comparisons are somewhat confounded due to their spatial separation, it is likely that Rio Grande silvery minnow would experience lower and more stable water temperatures if continuous streamflow was maintained downstream. However, during the summer of 2018, diel fluctuations of >10 °C and values that regularly exceeded 30 °C were observed during low-flow but continuous conditions (~0.03 $m^3$ $s^{-1}$) on the MRG between 3 and 10 km downstream of San Acacia Diversion Dam [51]. Although continuous flows alone likely do not provide thermal refuge, they can allow individuals the opportunity to seek refuge in upstream areas below diversion dams or near irrigation return outfalls, which may be cooler and have dissolved oxygen concentrations > 2.0 mg $L^{-1}$ [51], whereas the opportunity to do so during intermittency is eliminated. Although they are difficult to implement in a water-scarce basin, Rio Grande silvery minnow and other fishes in the MRG would benefit from development of water-use strategies that enhance connectivity among habitats during all seasons, allowing fish greater access to refuges during warm periods.

Relatively few Rio Grande silvery minnow were exposed to lethal water temperatures at the time of collection. Exposure to lethal water temperatures within isolated pools was likely limited because the majority of rescue efforts took place rapidly (within 24 h of drying) and during the early hours of the day before pool temperatures increased above critical thresholds. A slower response to new areas of drying or increased rates of channel drying would lead to more individuals being exposed to lethal temperatures. While the acute effects of water temperature on mortality are relatively well understood, the effects of chronic exposure to sublethal temperatures and rapid fluctuations in temperature on fishes is largely unknown [12,55]. We found that a large number of rescued fish experience both sublethal temperatures and relatively large temperature fluctuations. The effects of sublethal temperature exposure on Rio Grande silvery minnow have not been evaluated, but laboratory studies conducted on the regionally similar endangered loach minnow (*Tiaroga cobitis* Family: Cyprinidae) showed that sublethal static temperatures of 28 and 30 °C reduced growth rates compared to fish held at 25 °C [69]. These reduced growth rates suggest that fish experienced chronic stress at these thermal thresholds. Although lethal temperature tolerances have been evaluated for Rio Grande silvery minnow at varying life stages, it is unknown how chronic temperature stress affects this species.

The ability of an organism to respond to thermal change likely depends on the magnitude of the temperature shift, the frequency of thermal change, and the ability of individuals to acclimate to constant or fluctuating diurnal cycles [70]. While it is common for streams to experience daily fluctuations of ~4°C [12,71,72], our model results show that, on average, isolated pool temperatures fluctuated by >10 °C within a relatively short 8 to 12 h period. Our estimated diel fluxes are consistent with previous studies where temperature loggers were deployed in isolated pools within the MRG [51,60]. Growth rates in salmonids are reduced when subjected to sublethal, dynamic water temperature regimes [72,73], though these have focused on much lower temperature changes than what Rio Grande silvery minnow experience in isolated pools.

The observed dissolved oxygen values within the pools can largely be attributed to temperature controls of the solubility of oxygen in the water [68]. An inverse relationship between temperature and dissolved oxygen has been observed during short-term (3 days) deployments of high-frequency sensors within isolated pools in the MRG [51]. The reduction of biological controls on dissolved oxygen, via a reduction in rates of gross primary productivity and ecosystem respiration, has been observed in other rivers following pool isolation [74]. However, we observed that dissolved oxygen values elevated above the temperature–dissolved oxygen solubility curve indicate that inputs from

primary production [75] could be contributing, as considerable periphyton biomass was observed in a small number of pools during rescue and during summer low-flow conditions in the MRG [76]. The dissolved oxygen concentrations observed below the temperature–dissolved oxygen solubility curve may be a result of deep groundwater inputs within the MRG [77–79], upwelling of hyporheic waters [80], or high rates of heterotrophic metabolism within the hyporheic zone [81].

Regardless of the mechanism controlling diel dissolved oxygen dynamics, we observed that 496 pools that contained 367 juvenile and 1387 adult Rio Grande silvery minnow were <2.0 mg/L, whereas locations below diversion dams and irrigation return drains during low-flow conditions provided more suitable dissolved oxygen conditions for Rio Grande silvery minnow [51]. Although periods of acute hypoxia have also been observed within the MRG during periods of connectivity [51,82,83], periods of acute hypoxia in isolated pools have been observed in groundwater-dominated headwater streams [15,84]. However, native cyprinid species with low tolerance to hypoxia in the laboratory have been found to persist in isolated pools [15]. This variation in laboratory and field observations suggests that hypoxic conditions may not result in mortality of native species within isolated pools, but may favor non-native extremophile fishes [15]. Non-lethal effects of short-term hypoxia include transgenerational reproductive impairments [58], swimming performance, and behavior [85,86]. As a result, a higher threshold for sub-lethal effects of hypoxia on fish is recommended [87], and we propose that it should be further evaluated and implemented for Rio Grande silvery minnow.

Despite relatively few Rio Grande silvery minnow being exposed to elevated water temperatures, survival of rescued fish was context dependent. Survival was low in the weeks following capture and transport to a laboratory setting for fishes rescued in June through August. The markedly higher overall survival of Rio Grande silvery minnow collected under lotic conditions compared to isolated pools and lower survival of fishes collected in summer months demonstrates the cumulative stressors of confinement, temperature, and otherwise declining water quality conditions through the year during streamflow intermittency [51]. In both 2018 and 2020, almost no young-of-year fish were collected due to low spring runoff, resulting in failed recruitment [33]. Body condition, age, and season can all affect thermal tolerance in fish [88]. Possibly, adult fish are in poor condition after spawning and may have died regardless of exposure to high water temperatures. Future research should focus on both improving survival after rescue and examining differential effects on survival of adults and young-of-year. We also stress that these are likely conservative estimates of post-rescue survival: Fish were held in sterile, predator-free conditions, with optimal water quality, and were provided access to food resources.

As river drying occurs, it is likely that increased temperatures and reduced flows limit the ability of individuals to seek thermal refuge. Thus, it is likely that Rio Grande silvery minnow are exposed to lethal and sublethal temperatures for several days prior to stream intermittency and could be physiologically compromised before being rescued [51]. Handling and transport alone are significant stressors on healthy Rio Grande silvery minnow [89]. During rescue, mitigating stress to increase overall survival would prove difficult to implement. Fish transported in live wells mounted on the back of all-terrain vehicles cannot reasonably be acclimated to the broad daily temperature range from which Rio Grande silvery minnow are collected. The inability to properly acclimate fish during rescue along with additional handling stress may further limit the ability of individuals to survive once translocated. Water in holding tanks is generally around 23 °C early in the day and increases to about 27 °C at the time of release. Given the large number of pools that can be encountered daily (>100) and the long distances covered (>10 km), acclimating fish from individual pools would not be feasible. Rescuing a smaller number of Rio Grande silvery minnow per day with the intention of increasing post-rescue survival may be possible. However, this may be ineffective, as pool size does not necessarily predict numbers of stranded Rio Grande silvery minnow [37]. Rescuing fish prior to drying, while increasing survival, is also more difficult, as sampling efficiency is much lower compared to collecting from isolated pools. Further, this period of time overlaps with Rio Grande

silvery minnow spawning [33]. Handling of fish during that time could not only affect their ability to spawn but would not likely prevent their offspring from being stranded in isolated pools, as they produce neutrally-buoyant, non-adhesive eggs that passively disperse downstream [35,90].

Given that Rio Grande silvery minnow are a relatively short-lived species that rarely live longer than two years in the wild, the effects of streamflow intermittency on survival and growth warrant concern. Immediate effects of river drying include direct or indirect mortality, which could reduce population size. Further, little is known about the effects of chronic stress, including sublethal stress and its effects on reproduction. Increased mortality of young life stages or reproductive impairment could reduce demographic resiliency and eliminate reproductive contributions in subsequent years. Coupled with a 90% reduction in geographic range and fragmented habitats in the remaining population [26,90], streamflow intermittency further hinders recovery of Rio Grande silvery minnow, an outcome likely to be shared by many similar species with opportunistic life histories unless effective conservation actions are implemented [91,92].

Recovery of Rio Grande silvery minnow and other imperiled freshwater fishes will depend on proactive actions as opposed to reactive actions. Our results show that only a small portion of fishes rescued from isolated pools survive in the short term and that rescue is likely ineffective at mitigating the negative effects of streamflow intermittency on the population over the long term. For Rio Grande silvery minnow, rescue and translocation represent a conservation trap, in which concerted conservation efforts to offset species' declines result in actions that are perpetuated and unsustainable in the long term [93]. While fish rescue and translocation may be useful in dire conditions, their use as a regular conservation action to offset widespread, frequent streamflow intermittency has hindered recovery of Rio Grande silvery minnow. Rather than reactively rescuing stranded fish during intermittency, conservation actions need to examine ways to limit the ultimate causes of streamflow intermittency in order to achieve recovery goals. Such proactive efforts face multiple challenges in the MRG, chiefly declining precipitation [23,94] and surface flows [25,95] coupled with over-appropriation of water [96]. However, proactive actions are likely more effective than reactive actions. Proactive conservation actions may be less expensive over the long term and may prevent "conservation-reliant species" [93,97,98].

Climate change and increased human demands for freshwater will likely increase the frequency, duration, and spatial extent of streamflow intermittency, resulting in more fish stranding, greater fragmentation of habitats, and higher water temperatures. It is possible that similar rescue efforts will become more common in other systems. Thus, the effectiveness of rescue efforts must be evaluated. Population simulations may prove useful for modeling the overall population-level impact of fish rescue; however, the effect is likely minimal unless post-rescue survival can be increased. Arguably, the most prudent management action should be to maintain a known level of base flow that maintains suitable water quality, which, in turn, supports the survival of fishes and does not induce unnecessary stress. Recovery of threatened fishes will be difficult in the absence of proactive conservation efforts designed to promote self-sustaining populations.

## Appendix A

**Table A1.** Model parameters relating temperature of isolated pools or stream temperatures in a perennial section of the Rio Grande, New Mexico, to time of day and day of year in a linearized sinusoidal mixed-effects model.

| Model | Parameter | Estimate | SE | *t*-Value |
|---|---|---|---|---|
| | Isolate Pools | | | |
| | Intercept | 16.9809 | 1.2183 | 13.938 |
| | sine time | −5.1871 | 0.3277 | −15.83 |
| | cosine time | −8.495 | 0.8906 | −9.539 |
| | sine time (2nd harmonic) | 1.2147 | 0.2123 | 5.722 |
| | cosine time (2nd harmonic) | −1.537 | 0.3304 | −4.652 |
| | sine day | 6.1214 | 1.0853 | 5.64 |
| | cosine day | −5.3918 | 1.1174 | −4.825 |
| | sine day (2nd harmonic) | 4.4314 | 0.5364 | 8.262 |
| | cosine day (2nd harmonic) | −1.926 | 0.1974 | −9.758 |
| | Perennial Flow | | | |
| | Intercept | 14.17188 | 0.103947 | 136.34 |
| | sine time | −1.55107 | 0.00554 | −279.97 |
| | cosine time | −0.55226 | 0.005598 | −98.65 |
| | sine time (2nd harmonic) | 0.479119 | 0.005553 | 86.29 |
| | cosine time (2nd harmonic) | 0.169554 | 0.005589 | 30.34 |
| | sine day | −3.68166 | 0.005618 | −655.37 |
| | cosine day | −9.53207 | 0.005798 | −1643.96 |
| | sine day (2nd harmonic) | 1.024666 | 0.005507 | 186.05 |
| | cosine day (2nd harmonic) | −0.83944 | 0.005737 | −146.32 |

**Author Contributions:** Conceptualization, T.P.A. and J.K.R.; Methodology, T.P.A.; Software, T.P.A.; Validation, T.P.A., T.A.D., and J.K.R.; Formal Analysis, T.P.A.; Investigation, T.P.A., T.A.D., and J.K.R.; Resources, T.P.A.; Data Curation, T.P.A.; Writing—Original Draft Preparation, T.P.A.; Writing—Review and Editing, T.A.D. and J.K.R.; Visualization, T.P.A.; Supervision, T.P.A.; Project Administration, T.P.A.; Funding Acquisition, T.P.A. and J.K.R. All authors have read and agreed to the published version of the manuscript.

**Funding:** This research and APC was funded by the U.S. Bureau of Reclamation and U.S. Army Corps of Engineers.

**Acknowledgments:** We are grateful to W. Jason Remshardt, Tristan Austring, Judith Barkstedt, Matthew Nolan, Kjetil Henderson, and Rebecca Cook for their assistance with field efforts and data collection. We thank Nick Whiterod, Martinique Chavez, and two anonymous reviewers for providing constructive reviews that improved this manuscript. All fish were handled under U.S. Fish and Wildlife Service permit TE-676811-11 and New Mexico Game and Fish Department permit 1776. The views presented here reflect those of the authors and do not necessarily reflect the views of the U.S. Fish and Wildlife Service or the U.S. Army Corps of Engineers.

**Conflicts of Interest:** The authors declare no conflict of interest.

**Data Availability Statement:** Data used in this manuscript are available at http://dx.doi.org/10.17632/bz3dvhf95s.1.

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
