# Peer review of "Fish Rescue during Streamflow Intermittency May Not Be Effective for Conservation of Rio Grande Silvery Minnow"

_water, doi:10.3390/w12123371_

Round 1

Reviewer 1 Report

This study investigates the issue of increasing intermittency of water flow in a southwestern river. Removal and translocation effectiveness for an endangered minnow with respect to elevated water temperatures are  examined through field observation and an aquarium monitoring period. The study will be useful to those concerned with conservation decisions in southwestern rivers and stream habitats which are under increasing threat due to water-use conflicts and climate change.

Overall I found the study well written and thought provoking. There were however, several areas where the text will require reorganization and clarification before the work will be acceptable for publication. 

Major issues:

The introduction is lacking some important information, some of which appears later in the manuscript, other elements are missing at present. First, the system should be described in better detail. How large are the stagnant pools? Are they scattered about the sides of the river bed as small oxbow features perhaps? Does the entire river cease to flow, with these pools being only the deepest parts of the channel? Does the river somehow flow in sections upstream and downstream of this area, but not actually through it? The number and duration of persistence of these pools are important to establish in the introduction for readers not familiar with this area. 

The second issue that needs better introduction is a description of the basic life history of the Rio Grande silvery minnow. Many important aspects of this species biology are brought up in various spots of the Discussion, but they should be established in the introduction since the study design only makes sense in the context of the fishes life cycle. A brief paragraph describing the relevant biology, trophic role, lifespan, spawning season, historical extent would suffice. 

Another element missing from the Introduction is a description of the existing rescue program and logistics. Some but not all of the issues are touched upon in the Discussion, but it really needs to come in the introduction. Again, the mechanics of rescue, triggering conditions, and temperatures and durations of various stages in the rescue are critical to understanding whether or not the study design was appropriate. How much handling and jostling are fish subjected to during relocation? Are they merely moved upriver or downriver to a location where flows are active? Would 100% of the fish die if they weren't moved? This information should come in the Introduction and is needed to determine how well the lab holding conditions mimic those of actual translocation. 

Lastly, the Discussion is in need of some reorganization. I was generally disappointed with the lack of reference and Discussion of other studies that may provide supporting findings, comparable results, or that have addressed similar issues. This is needed to broaden the applicability and transferability of the findings from this minnow to other fishes in other systems.

Detailed comments by line number:

2. The title should have some reference to relocation of fish since that was a focus of the paper.

11. "variety" implies that a few different examples will be given. 

13. "rescue" means different things, best to define it here as "relocation to suitable habitat" here, or at least at the first mention in the Introduction.

16. Switch from "imperiled" to "endangered" (if thats correct) which as more specific meaning.

22. A brief phrase explaining "conservation trap" is needed.

31. "abstraction" should be "extraction" it seems. and on line 54

33. What about the often related and compounding issue of low dissolved oxygen with increasing temperature? The sole focus on temp without mentioning any other, even related, issues is a bit narrow. What effects of no flow on their food items? What do they eat anyway?

42. This sentence belongs with previous paragraph.

67. Clarify this sentence. Disconnect between "allowing persistence", and "lowering mortality"

71. Rescue needs to be defined. It seems most appropriate to change this to "relocation" based on what is apparently involved as described in the Discussion. Rescue can just mean the part where fish are removed from the pool of thermal demise.

77. Ecological benefits are not really a consideration here, though. The primary motivation if this is a threatened species is simple survival. 

78. Higher than what?

80. Here is another case where it would be good to know where they are being translocated to? Another river or stream, a different part of the Rio Grande?

101. Again a strange and singular focus on only temperature, rather than more broadly on water quality issues including O2. 

102. Please explain the initial circumstances and options considered around the decisions to begin relocation in the first place.

119. Clarify that this is the only season when intermittent flow is an issue. 

121. Clarify what the typical extent and duration of intermittency events may be. 

131. Material in this paragraph needs to be reorganized.

150. Clarify "each water temperature dataset".

153. Here is one place where it becomes necessary to have explained what is meant by "rescue" and what is involved. What actions are being simulated in the lab? Is the idea to replicate moving fish from a stagnant pool to a flowing stream after handling them for a time? Is the idea to house them temporarily with the goal of putting them back later? or merely to save them from death by keeping them in an aquarium setting?

155. Both lotic conditions? 3 groups? 9 groups? Clarify.

157. How were they transported and acclimated to the tanks? Handling often causes physical trauma (body and fin abrasions), stress, and increased oxygen use. Temperature and chemical shocks are likely if not done gradually and carefully. The transport alone may cause sublethal damage that opens routes for infection, were transport times to the lab and during field relocation the same? How about crowding? Disease? Any effort to cull those individuals that already looked stressed and unhealthy and could be disease sources. Most importantly, How does the transport and conditions in the aquarium replicate the relocation that is occurring in the field?

158. Aeration. Were O2 levels representative of those likely to be experienced at relocation sites in the field?

159. What is the comparison between city water and water from the transplant locations? Were there any concerns with the water chemistry or pH of the city water? Could it be that even with filtration the city water was less suitable for this stream fish and confounded the results, was all chlorine and other additives confirmed removed?

162. 19-23 C water in the lab is apparently too cold based on figure 3B. This should have been the same temp as the fish would experience in relocation. 

163. Clarify if this is typical and acceptable food to keep this species nutritionally healthy. Again, is food quality an issue in captivity like this versus in the translocated rescue scenario in a natural system?

167. This is an impressive number of samples. It would be good to convey if these are representative of all pools, or if this sample is biased in any way. 

168. Why are some samples below this temperature? also, clarify why rescues are done even when water temp is so low?

169. Looks like 18:00 is higher than 13, so the thermal stress is even longer in duration.

169. Reporting the basic proportion of pools over the various thresholds (and other statistics for the samples pools) would be informative.

175. Be sure to explain that perennial refers to "flow".

177. A few curious things about this figure, the rapid decline in the number of samples after 15:00 should be explained, as well as the lack of any samples in the 19:00 hour. Also the lack of low temps in the 7-8 interval. 

180. Caption is cut off. 

186. Clarify that these were "rescued" because the water temperature would have likely increased above their thermal tolerance later in the day. If that is the case.

188. Why were these not all dead if water temp was already above lethal max? Due to the 50% mortality used to estimate thermal max perhaps?

196. Why were they rescued if water was flowing? Due to high temps alone? If that is the case, I'm not sure that any restoration of flow would work to solve this problem. 

198. Much higher! perhaps express as a factor increase (e.g. 3 to 20 times higher).

205. I wonder if you don't need some control groups in this case. You'd also need to capture, handle, and transport a group of fish that were in no danger of prior thermal stress to see how many survived. What would mortality have been had you merely captured fish at these times that were not in need of rescue? Could it be that fish in summer are just in generally less healthy condition? I'm not sure that this design is adequately addressing either the handling issue, or reflecting the conditions when fish are relocated in the field.

220. Important points are being made here, but perhaps belong better in a separate section of the discussion focused on management recommendations.

227. Stagnation a better word? "drying" suggests to me that there is no water in the pool. 

243. Reword "cyclic cycles".

248. 60 d?

249. Clarify, are you suggesting the aquaria are more natural?

261. This is another example where it would be good to have had a bit more general life history explained earlier. How many years do they live, when do they spawn, etc..

265. It is not clear to me that these were actually better conditions than had they been quickly released in natural habitats. 

276. This is the first time we've heard that they are rescued by relocating them upstream to areas of perennial flow. This needs to be explained earlier.

280. Is the stress of relocation to another river segment the same or similar to the stress of moving them to the lab? Are distance, jostling, and handling time all the same? This could affect the results with more mortality occurring the longer they are handled. 

283. What would this do? Decrease the handling and storage time?

285. Please clarify this sentence.

301. Need to clarify, probably in the intro that mortality would be 100% if fish were not rescued. Even so, 10% is better than none surviving for a threatened species and may be worth it. 

312. Some repetition with earlier material in this paragraph.

321. If fish are in need of rescue even before the water stops flowing it is unclear what rescuing them does at all, unless they can be moved to a place with cooler water that is still flowing. Climate change may make this system simply unsurvivable. i.e., even if flows are ok, are temps too high anyway?

Reviewer 2 Report

I have now finished reviewing the article “Water, or lack thereof: streamflow intermittency hinders conservation and recovery of an imperiled fish”. I find it a very nice study and very much like the applied aspect of the research and the conservation cues that could potentially emerge from it. The severe streamflow intermittency may strongly impact species survival. Especially concerning the continuous and increasingly rapid impact of climate changes on a great variety of environments, urgent measures should be taken to rescue and translocate imperiled animals but, overall, to a re-rethinking of conservation strategies. Because of this, I believe this paper meets the purposes of the journal, but there are some details about the approach that the authors report only scratching the surface of what could be really discussed.

This is why I suggest to re-think some concerns arising from the study and reported in my comments here below and, if possible, add these points to the discussion.

Here below a few major comments:

Title.

1- The title is too generalized while the paper  specifically focus on the Rio Grande silvery minnow. I suggest to report the name of the specie within the title.

Introduction.

1- Line 57-59. Please add in this section a few more details about the species of interest (expected/known survival rates in different conditions, biological cycles, diet, habits). This could help readers to immediately focus on the potential risks that could affect this species.

2- Lines 50-54. Here the authors report a few examples of other species that could be strongly impacted by streamflow intermittency. It is therefore to be noted that different species, with completely different sizes or status assessments, could require completely different conservation measures or may be affected differently by the streamflow exchanges described in the paper. As you cited these species in the Introduction, a possible outline and discussion of these aspects are completely lacking in the discussion.

3- Lines 82-88. This part should be moved to the discussion section and compared to what observed by the authors.

Material and Methods.

1- Lines 153-165. After rescuing fish, the animals were maintained in controlled conditions. It is not fully clear how the authors excluded that mortality after rescue occurred due to past high-temperature exposure. How they estimate the fish state of health at the time of rescue (a few parameters have been taken)? How they can exclude that mortality can be occurred due to induced stress due to a rapid change in environmental conditions. This is not clear.

Discussion.

1- I found this particularly long and in some parts a little bit dispersed and repetitive of aspects already raised in the introduction. I suggest shortening the discussion and revising the story-telling trying to deeper the discussion with some clear authors' suggestions about possible conservation guidelines.

2- See point 2 of the introduction.

3- Line 288-292. Considering the high demographic resilience and the very short life span with high reproductive efforts characteristic of the life cycle of this species, what does adaptation to severe climate conditions? It is indeed presumed that several generations had already been strongly impacted by streamflow intermittency and exposure to extreme environmental conditions. Have the authors taken into account a possible revealing of signals of adaptation extrapolated by their data that have been collected across different years? I think that a paragraph with a note of caution should be introduced.

Round 2

Reviewer 2 Report

I have presently reconsidered the manuscript in this revised version and I think authors have fixed all raised points successfully.

Thus in my opinion the manuscript can be accepted in the present form.